# Cemeteries as a Part of Green Infrastructure and Tourism

**Ágnes Sallay \*, Zsuzsanna Mikházi \*, Imola Gecséné Tar and Katalin Takács**

Institute of Landscape Architecture, Urban Planning and Garden Art (TTDI), Hungarian University of Agriculture and Life Sciences (MATE), 1118 Budapest, Hungary; gecsene.tar.imola.csilla@uni-mate.hu (I.G.T.); takacs.katalin@uni-mate.hu (K.T.)

\* Correspondence: sallay.agnes@uni-mate.hu (Á.S.); zsuzsanna.mikhazi@gmail.com (ZS.M.)

**Abstract:** The world's population and the proportion of it living in cities and urban areas has exploded in recent decades. In the European Union, 62% of the population lives in urban areas and 80% in suburban areas, and these proportions are projected to increase further in the coming decades. It has long been researched and proven that 'urban greenery' can play a major role in mitigating the so-called urban heat island effect, and during the COVID-19 pandemic the role of daily recreation has come to the forefront. The combined memorial, recreational, and touristic use of cemeteries can help to ensure their economic management, and thus the long-term preservation of their value. In international tourism the model of managing cemeteries as tourist attractions already exists; however, this is not yet part of conventional practice. In addition to traditional cemetery tourism (e.g., visiting the graves of celebrities or enjoying artistic treasures and values), cemeteries are used as venues for events and sports activities. In Western Europe forest and park cemeteries have been established since the 19th century, and their large green areas and open spaces are a prerequisite for their use as public parks. Thus, the use of cemeteries as public parks is a common if quite specific practice. Our aim with this article is to identify the green space values of Budapest's cemeteries, in addition to their well-known cultural and architectural significance, as well as to define the potential and means of their involvement in tourism-related activities. Another aim of our study is to raise awareness of green cemeteries within the tourism profession as potentially wider tourist attractions. We consider it important to draw the attention of decision-makers to the significance of the greenspace values when preserving or reusing closed cemeteries. Based on our work, other major cities in Hungary can identify and exploit the touristic and green space potential of their cemeteries.

**Keywords:** cemetery; tourist attraction; green space; urban green infrastructure; cemetery tourism

## 1. Introduction

The role of urban green infrastructure or urban green areas is growing in cities as a result of population growth and urban intensification. In recent years, the COVID-19 pandemic has amplified the importance of open spaces for the local population, who spend more time in parks and other green spaces available in close proximity to their home. Cemeteries, similar to urban parks, represent an important part of the urban ecosystem, being the remaining semi-natural habitat of many species of plants and animals. However, there is no equivalence between cemeteries and public parks: while their functions and uses are similar in many respects, specificity could be mentioned as well. Cemeteries are part of the urban green infrastructure network, however, their use is limited in both time and in function. Although the use of cemeteries and parks has converged in recent decades, significant differences nevertheless exist between them (Figure 1).

While the use of cemeteries for recreation and tourism purposes is not yet considered conventional, the rise of 'green needs' can be expected to increase their role in the future. On average, four million people visit Budapest cemeteries every year, and on the All Saints' and All Souls' Day the number of visitors can reach 1–1.5 million individuals over 2–3 days.

However, no general statistics are available on the number of visitors coming to cemeteries with targeted touristic and recreational aims.

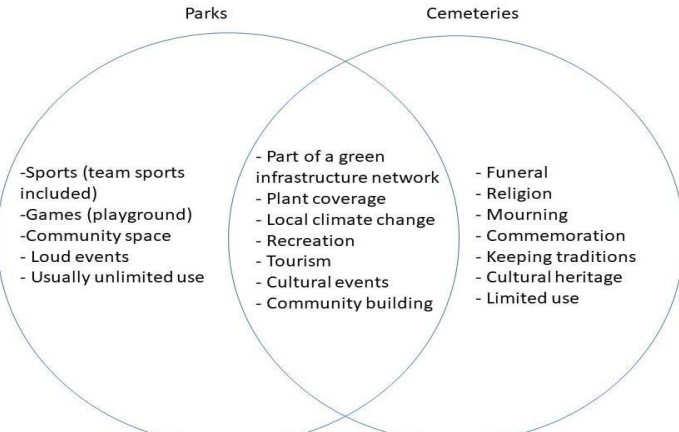

**Figure 1.** Differences between the functions of cemeteries and public parks (edited by the authors).

The value of cemeteries for the tourism sector lies in both the cultural aspects they offer and in the vegetation and green spaces that frame their physical realities, which can offer many services. "There are many types of cemeteries. As many as man and his follies," writes Károly Eötvös [1] (p. 175). The layout, the use of materials, and the way of commemoration can vary from simple wooden headstones to massive stone carvings and sculptures. In Hungary, there are several legendary cemeteries where families commemorate their deceased relatives according to special traditions, and these sites have become tourist attractions. In Szatmárcseke (HU), the graveyard with its nearly 600 dark headstones representing a man's head or a man lying in a boat is a unique sight [2]. In Balatonudvari, near the national road crossing the village, the local cemetery is protected due to its heart-shaped tombstones carved in white limestone [3] (Figure 2). The cemeteries in the region of Őrség (a cultural landscape area in Hungary near the Slovenian border) are a reminder of a unique burial tradition in the country. In Kercaszomor and other settlements in this region there exist some old and unique wooden headstones that were once erected for Calvinist deceased. This tradition is now extinct, and cemeteries with these sights are today only a tourist attraction [4]. In Hajdúböszörmény, in the former Calvinist graveyard known as the Nyugati Cemetery, we can see typical 19th century gravestones with their boat-shaped headstones [5].

In addition to cemeteries that are special due to their burial traditions, there are other tourist attractions related to memorial places in the country: the Historical Memorial Site of the Mohács Battle, the Budaörs Military Cemetery and Peace Park (Deutsch-Ungarischer Soldatenfriedhof) [6] (Figure 3), the British Military Cemetery in Solymár, the Turkish memorial tomb of Gül Baba in Budapest (Gül baba türbéje) [7], and the Old Christian burial chambers site in Pécs, which is on the Unesco World Heritage List [8].

### 1.1. Theoretical Background

In previous works, the authors have dealt with religious tourism [9,10] and the development, structure, and history of cemeteries [11–13]. This study now focuses on the role and potential of cemeteries for recreation and tourism. To this end, we conducted a literature review on cemeteries and recreation and cemeteries and tourism.

### 1.1.1. Cemeteries and Recreation

"Today, cemeteries are more than a place of reflection. They are a place of beauty and a place of history" [14]. A cemetery is a green open area [15], a "garden" with architectural and sculptural elements. It performs an ecological function and it is a permanent element of the landscape. It offers a chance for survival to many species of plants and birds, especially in cities, and natural "monuments" are often found among the many trees [16]. Because of

their characteristics and location, throughout history cemeteries have often had a secondary function in addition to their primary one [17,18]. In the Middle Ages, church graveyards were often the central points in cities, and were the sites of fairs and festivities as well as being used for local parliaments, trials, preaching, miracle play performances, folk rites, executions, and demonstrations; however, the cemetery has always been a place of solemnity.

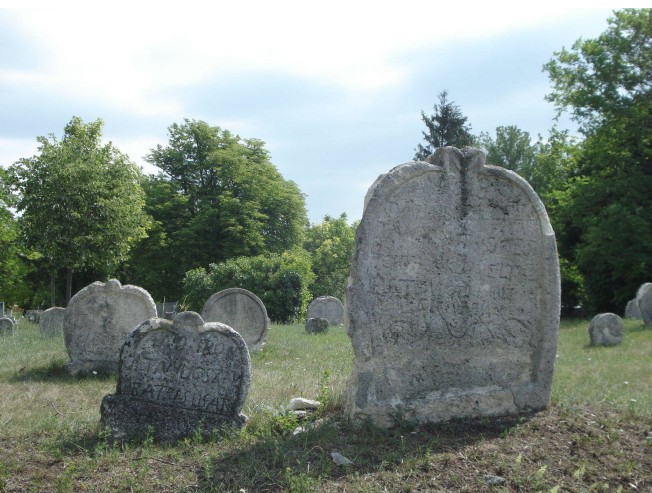

**Figure 2.** Heart-shaped gravestones in the cemetery of Balatonudvari. (Photo: Imola Gecséné Tar, 2016).

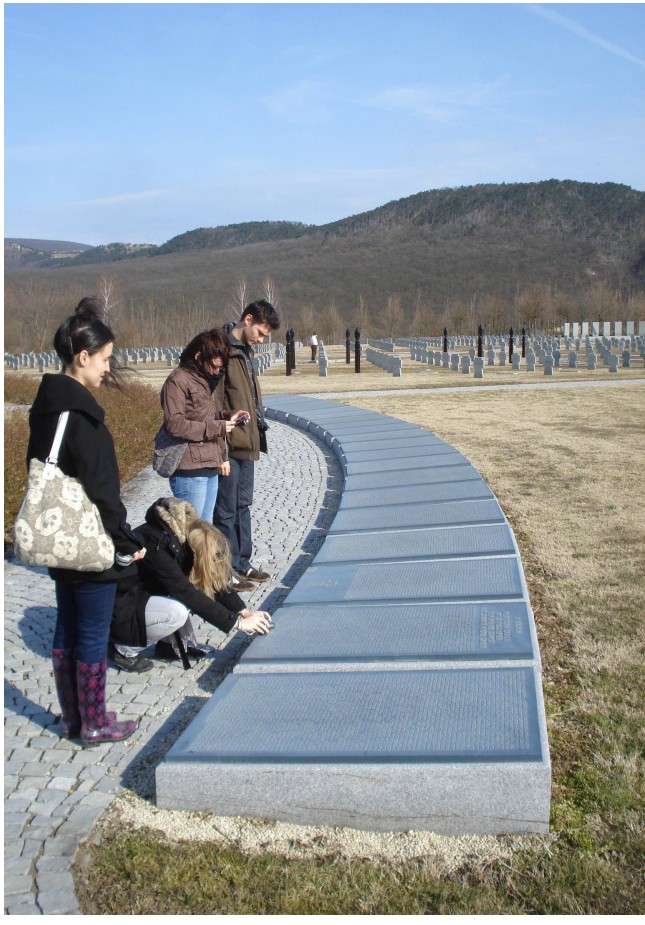

**Figure 3.** Tourists in the German–Hungarian Military Cemetery in Budaörs. (Photo: Imola Gecséné Tar, 2018).

The majority of historical cemetery complexes are park-type areas, and are endowed with recreational facilities: clean air, silence, limited urbanization, aesthetic landscape features, and favorable climatic and bioclimatic conditions [16]. Visiting cemeteries can therefore be an opportunity for recreation. "It can be a place to get one's thoughts rested and let them stretch themselves out. So, it is very good mentally. Yes, good to the eye and good for the head." (man in his 40s visiting the Old Town Cemetery) [17] (p. 1). As cities become denser, green spaces are in danger of decreasing. Evensen et al. [19] (p. 76) argue that "in densified parts of cities the cemetery may be the closest greenspace accessible for every-day use" [20] (p. 2). This may have consequences for how urban cemeteries shift from being burial spaces to becoming spaces for recreation [17,18].

### 1.1.2. Cemeteries and Tourism

In the twentieth century, people began to travel to sites associated with death out of curiosity rather than because of their philosophical or spiritual connotations, thus initiating the origins of dark tourism. Dark tourism is the act of travel and visitation to sites, attractions, and exhibitions that have a connection to real or recreated death, suffering, or the seemingly macabre as a main theme. Tourist visits to former battlefields, slavery-heritage attractions, prisons, cemeteries, particular museum exhibitions, Holocaust sites, and disaster locations all constitute the broad realm of 'dark tourism' [21].

Cemetery tourism (thanatourism) is a specific sub-section of dark tourism that is becoming increasingly popular [22]. Tourists wander through burial grounds with the aim of discovering the artistic, architectural, historical, and scenic heritage that often abounds in cemeteries. The changing perception of cemeteries from a place for burial towards a cultural heritage space provides several opportunities for tourism. It enables the community to explore the development of products and services that help the destination to gain new income while preserving its heritage [23].

Cemeteries are more than just resting places for the dead; they serve a practical purpose, serve as historical markers, reflect cultural values, and impress visitors with their gorgeous designs and much more. Many people find cemeteries particularly interesting for these and other various reasons. Most cemeteries welcome the public free of charge, and many offer thematic maps, brochures, smartphone apps, audio tours, or guided tours that highlight notable graves, statues, monuments, chapels and other architectural structures of the site [14]. There are many books that have been written on the topic in recent years, for example, Stories in Stone: A Field Guide to Cemetery Symbolism and Iconography, by Douglas Keister; Beautiful Death: Art of the Cemetery, by David Robinson and Dean Koontz; and Your Guide to Cemetery Research, by Sharon DeBartolo Carmack. Today, many tourist guide books include cemeteries which may attract different groups of travelers and inspire the development of newer cemeteries.

One of the 45 European Cultural Routes of the Council of Europe is the European Cemeteries Route, certified in 2010. The European Cemeteries Route refers to cemeteries as 'places of life', environments that, as urban spaces, are directly linked to the history and culture of the community to which they belong and where people can find many of their references [24].

### 1.1.3. Summary of the Literature Search

Based on the literature review, we have summarized the possible functions of cemeteries, which are shown in Figure 4. Many people see cemeteries as somber places with little connection to the local environment and community; however, we believe this could not be further from the truth. Cemeteries bring families together and provide insight into local history.

We have analyzed in detail the role that cemeteries can play and the needs they can meet for both local people and tourists. The basic and primary function is burial and commemoration. All other functions can only be carried out with this in mind and subordinate to it. The secondary function is linked to the recreational needs of the local

population through the preservation of local natural and architectural assets. The tertiary function is to satisfy the needs of tourist visits.

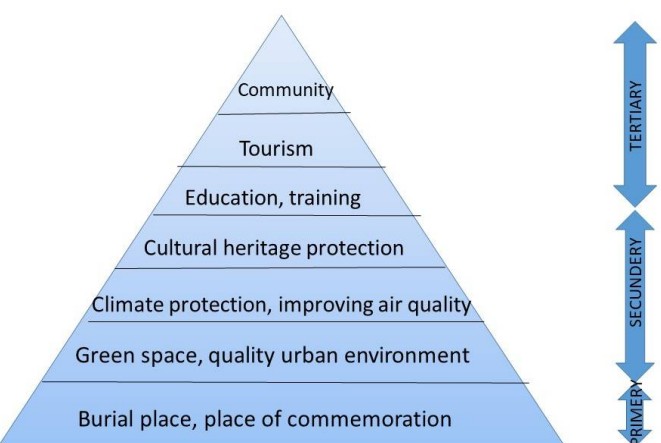

**Figure 4.** Diverse functions of cemeteries (edited by the authors).

*1.2. Presentation of the Sample Area: Green Infrastructure and Cemeteries of Budapest*

In the Middle Ages, the greenspace areas of Pest and Buda developed in a similar way to those of other European cities. Within the castle walls and the city-enclosing walls, castle gardens, manorial gardens, and many small kitchen gardens were erected, while outside the walls the large meadow fields and forests enriched the landscape. The medieval cemeteries were located within the town, around the churches.

Hungary's development was severely set back by a century and a half of Turkish occupation during the 16–17th centuries. Despite this, Pest and Buda developed gradually, although in different ways. Within the walls of Buda's castle the free open spaces had been used up, and both the expansion of other urban areas and property rights were regulated according to political and military defense. In contrast, after the Turkish dominance there was more intensive development at Pest, with construction and a greater pace of transformation, although at that time civil governance was far away from creating public green spaces or parks [25]. In the area of today's Budapest the practice of burial around churches had ceased after the Turkish conquest, and cemeteries similar to the present ones started to be developed in different parts of the city.

In the 18th century, due to the large number of newcomers the development of the suburbs around Pest began. At that time, few public spaces were created in the densely built-up urban areas. With the advance of civilisation, the demand for public green spaces rose, just as in other European cities. As more and more city dwellers became increasingly detached from their peasant life, they were keen for recreation and leisure. Thus, the city mayor proposed the creation of a new forestry zone, which later became the first public park in the capital (today's Városliget), as well as a promenade (Városligeti allee) connecting the urban areas to the new recreational zone. This provided a pleasant open space for the entertainment of the citizens of Pest and its surroundings. In Buda the royal gardens were open to the public, although on a limited basis, and new pleasure gardens (Várasmajor, Horváth garden) were created in the late 18th century. In addition to gardens, the linear axes, alleys, promenades, and walkways played an important role in the early days of the development of public open greenspace [25,26].

The regulations on the removal of cemeteries issued in 1777 led to the opening of several new cemeteries in Buda and Pest. Two new cemeteries were opened in Buda, though both were closed at the end of the 19th century [27] (pp. 4–5) [28] (p. 235). In Óbuda, the cemetery surrounding the parish church ceased to exist in 1744. Instead, two new cemeteries were opened in 1780, along with one for the Jewish population of Óbuda [27] (p. 7). In the 18th century there were several cemeteries in Pest, and from the second half of the 18th century onwards new cemeteries were established outside the city walls. In

the 1790s the first central cemetery in Pest, the Váci Road cemetery, was opened; it was eliminated around 1910. There was a Jewish cemetery next to it as well [27] (pp. 8–10).

In 1846 the Pest City Council decided to replace the old cemeteries with a large central cemetery. The Kerepesi cemetery (now National Graveyard on Fiumei Road) was opened on 15 June 1847. In 1885 the entire ensemble was declared an ornamental graveyard. In 1874, the City Council handed over a part of the Kerepesi cemetery to the Israelite community. Burials took place in the separate Jewish cemetery section in Salgótarjáni Street until 1950, and it has been preserved as a closed protected cemetery. In 1972 a tender was launched for the arrangement of the cemetery and the landscaping process, which is ongoing today. Today this cemetery has a special status as a national pantheon, although it continues to be possible to carry out public burials there [27] (pp. 10, 14, 16, 31), [28] (pp. 235–236, 238–239).

The Parliament of 1872 voted for the unification of Pest, Buda, Óbuda, and Margaret Island. The newly unified capital sacrificed for the development of its metropolitan character, following the European (Viennese and Parisien) models by removing its rural character by building new residential areas, representative axes, squares, and public institutions. In 1873, the afforestation of Gellért Hill was started. After the construction of the Chain Bridge, the Buda Hills as green open spaces have become accessible to the citizens of Pest and increasingly popular as a recreational destination. On the other hand, the hillsides were gradually urbanized by new villa areas, replacing traditional orchards and vineyards damaged in the Phylloxera epidemic [25].

Several new large cemeteries were opened in the merged capital. In Buda, the Németvölgyi cemetery was opened in 1885; it was intended to be an ornamental graveyard for all time, similar to the Kerepesi cemetery. Despite this intention, the cemetery operated for an unusually short period. The Németvölgyi Orthodox Jewish cemetery, which was opened in 1890 and used until 1961, stands on a small plot of land surrounded by a high fence. In 1894, the Farkasréti cemetery was opened in Buda, which is still in use today; it is the resting place of many famous Hungarian people. The Farkasréti Jewish cemetery is wedged between the two parts of the Christian cemetery [27] (pp. 5–6, 53, 58). A new public cemetery and a new Israelite cemetery were opened in Óbuda; however, these have now been closed down [27] (p. 7). In Pest, the expansion of the Kerepesi cemetery became impossible with the opening of the Jewish cemetery on Salgótarjáni Road, and a site for the new public cemetery had to be found elsewhere. On 1 May 1886 the New Public Cemetery of Rákoskeresztúr, Hungary's largest cemetery at 207 hectares, was opened. In 1891 the Kozma Street Jewish cemetery, Hungary's largest Jewish graveyard, was established next to the New Public cemetery, with the Rákoskeresztúr Orthodox Jewish cemetery adjoining it on the northern side [27] (pp. 32, 35, 44) [28] (p. 236).

The Fortepan photo collection contains photographs from the beginning of the 20th century until the change from the Soviet regime in Hungary in 1989, thus, the cemeteries in Budapest have been photographed beginning in the early 1900s. When we reviewed these photos, we found that the oldest photo analyzed was taken in 1900 in the National Graveyard, formerly called as Kerepesi Cemetery. There is a difference in the use of the two cemeteries from the very beginning: in case of the National Graveyard, the archive contains almost exclusively pictures of the burials and graves of famous people, while in the Farkasréti Cemetery it is the funeral motifs that appear in large number, of which the most striking is a horse-drawn hearse. For both cemeteries there are almost no photographs that do not show some form of vegetation, usually tree alleys or massive woody vegetation in the background. In the case of the Farkasréti cemetery, there was one photograph where the spatial structure and alleys were clearly recognizable (see Figure 5) in addition to the main motif of the funeral process.

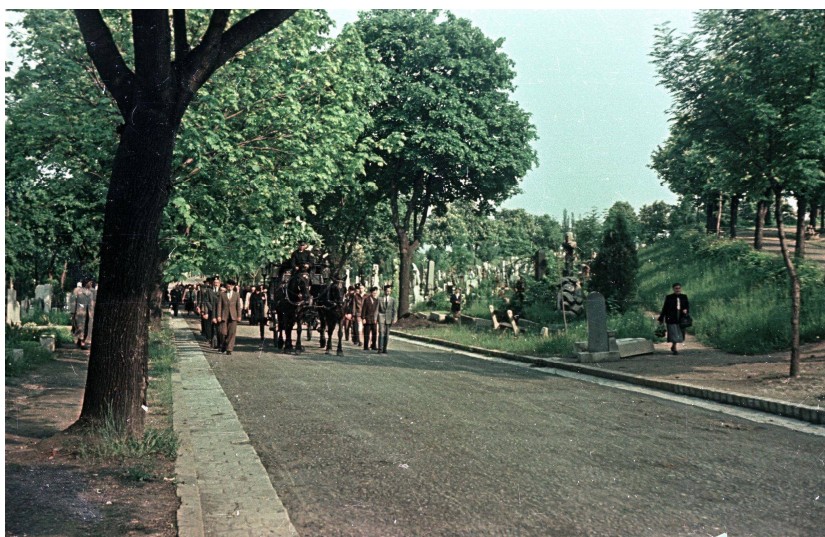

**Figure 5.** Alleys such as this one define the structure of the Farkasréti cemetery (Photo: Fortepan/Horváth Miklós dr, ID N°129338, Date: 1955).

At the turn of the 19–20th century the function and use of public parks evolved, with crowds from various social levels, coming to walk, play skate, and seek active or passive outdoor recreation. At this time, the general condition, maintenance, appropriate use, and thus popularity of the green spaces in the capital were all outstanding. Many public parks as well as the largest urban parks were enriched and transformed with newly integrated functions such as early childrens' playgrounds [25]. In 1910 the Óbuda cemetery, which is still in use today, was opened, and a Jewish cemetery was opened next to it in 1922 [27] (p. 68). In 1919 the Budapest Municipal Cemetery Institute (Budapest Székesfővárosi Községi Temetkezési Intézet), the predecessor of the existing Budapest Funeral Institute (Budapesti Temetkezési Intézet, BTI) began its operations. From 1949 onward, the cemeteries previously owned by the Catholic church were taken over by the capital [28] (pp. 236, 239).

Budapest gained its current extension in 1950 when several surrounding small villages were annexed to the capital and the present-day outlying districts were formed. The previously independent municipalities had their own cemeteries; therefore, in 1950 Budapest possessed 87 cemeteries [28] (p. 236). In the 1950s, several small cemeteries were closed. Most were liquidated, although some cemeteries were reopened in the second half of the 20th century; in some cases, urn cemeteries were built on their sites. Those former village cemeteries that were suitable for expansion remain in operation today [12].

After the Second World War and from the 1950s onwards, development and settlement of housing estates began in earnest, along with the creation of significant public greenspace. At that time, these residential areas had a wide range of functions as public green spaces and a very high green surface ratio. On the other hand, the number of touristic developments in the urban forests increased dramatically [29]. After the change of regime in 1989 the structure of the city became more dense and the role of green spaces, including cemeteries, became more important, mainly due to the loss of former industrial areas and their conversion into office buildings and housing estates (Table 1).

Today, there are fifteen public cemeteries and four Jewish cemeteries in use in Budapest (Figure 6, Table A1). The public cemeteries are managed by the Budapest Funeral Institute, with the exception of the National Graveyard on Fiumei Road (Kerepesi cemetery). Due to its special status, this latter has been managed by the National Heritage Institute since 2016. The operating Israelite cemeteries are managed under ecclesiastical care. The closed cemeteries are partly municipality-owned and partly church-owned, and their management is mostly unresolved.

**Table 1.** The changing role of cemeteries and other urban green spaces (by authors, 2021).

| Period | Urban Green Areas | Cemeteries |
|---|---|---|
| until the end of the Turkish occupation | small vegetable gardens, private residential gardens, extended agricultural land | cemeteries next to churches, burial places |
| 1700s | increasing density of urban development leading to an emerging need for public green spaces and parks | cemeteries appear on the outskirts of the settlement, independently of churches |
| 1800s | first public parks, large boulevards with alleys, afforestation, public squares and pedestrian areas with numerous trees | large new cemeteries established on the outskirts of the city. with the burial function remaining primary |
| 1872–1945 | more leisure activities in public parks; the first public playgrounds | large new cemeteries opened on the outskirts of the city due to rapid urban development, in which the graves of famous people become urban pilgrimage sites. |
| 1945–1990 | in parallel with the construction of housing estates, public green spaces around them are created; the first protected green areas are designated in the city, and park woodlands become significant | former village cemeteries around Budapest annexed to the capital by the creation of peripheral areas (many of which have been dismantled, although a few remain in use today), either as extensions or as urn cemeteries; cemeteries popular for recreational purposes thanks to attractive mature vegetation |
| 1990s | new residential and office developments created on brownfield sites, increased building density, growing demand for open greenspace | tourist use of cemeteries, mainly guided walks around the graves of famous people and tertiary usage such as classical music concerts and exhibitions |
| 2020 | pandemic closures have pushed people even more towards urban green spaces, for which demand has increased, and many have become overused | during lockdowns, people spend part of their recreational time in cemeteries, and mourning also 'attracts' more and more people to cemeteries |

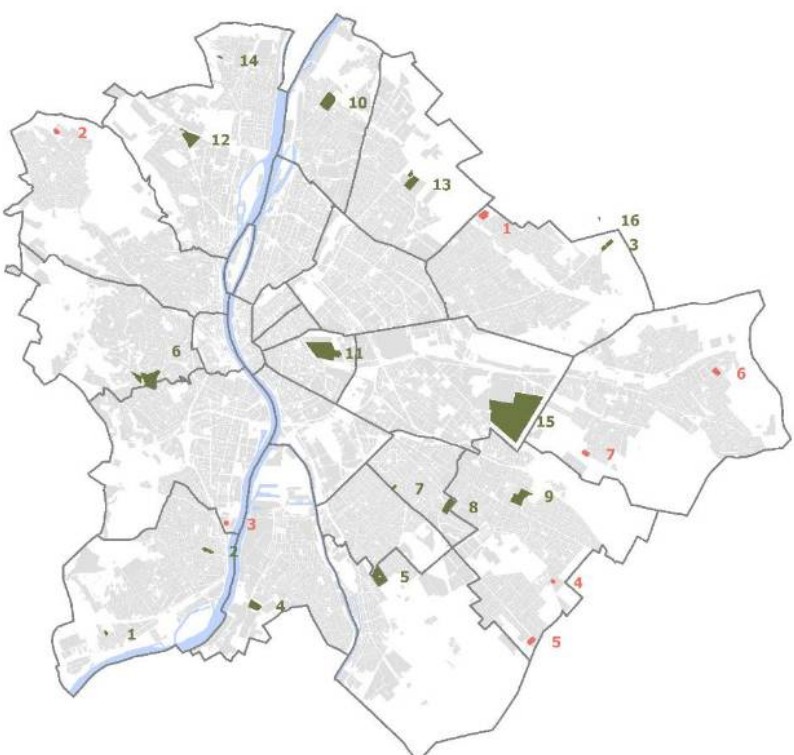

**Figure 6.** Overview map of functioning cemeteries in Budapest. Cemeteries managed by BTI (Budapest Funeral Institute) [30]. Functioning cemeteries (green): 1. Angeli Road Urn Cemetery, 2. Cemetery in Budafok; 3. Cemetery in Cinkota; 4. Cemetery in Csepel; 5. Cemetery in Pesterzsébet;

6. Farkasréti Cemetery; 7. Old Cemetery in Kispest; 8. Cemetery in Kispest; 9. Cemetery in Pest-szentlőrinc; 10. Megyeri Cemetery; 11. National Graveyard on Fiumei Road (from 2016 managed by NÖRI); 12. Óbudai Cemetery; 13. Cemetery in Rákospalota; 14. Tamás Street Urn Cemetery; 15. New Public Cemetery in Rákoskeresztúr; 16. Cemetery in Csömör (outside the administrative boundaries of Budapest). Closed cemeteries (red): 1. Cemetery in Rákosszentmihály; 2. Véka Street Cemetery; 3. Cemetery in Albertfalva; 4. Ganz Street Cemetery; 5. Nagykőrösi Road Cemetery; 6. Göcsej Street Cemetery; 7. Bocskai Street Cemetery.

*1.3. Green Infrastructure Network of Budapest and the Role of Cemeteries in the Capital Today*

In Budapest, as in other major European cities, the effects of climate change are increasingly noticeable and are expected to intensify in the coming decades. Forecasts indicate an increase in the number of heatwave days and tropical nights, which will make everyday life much more difficult for city residents [31].

The so-called urban heat island effect is a microclimatic phenomenon in large cities, where the temperature in built-up urban areas is significantly higher than in the suburban and rural areas surrounding the city. The causes of this phenomenon are complex; the most important are:

- lack of natural evaporative surfaces
- concrete and asphalt surfaces in cities absorb more solar radiation than they reflect
- vertical surfaces increase the absorption of radiation
- human activity generates heat in many different ways
- pollutants greatly modify the atmosphere
- changes to meteorological conditions such as wind direction, humidity, visible sky, precipitation, and radiation conditions [32–34].

Around 65% of Budapest's territory (34 thousand ha) is covered by green space and vegetation, and almost 2% of the city's territory is parkland; 40% of these green areas are managed by the Municipality of Budapest (420 ha), while the rest are owned or maintained by the district municipalities. Nearly 6000 ha, or 11% of the city's territory, is forested. The distribution of green spaces in Budapest is not even; in some inner-city districts (VI, VII) there is less than 1 $m^2$ of public park per inhabitant, while in the suburban districts there is good coverage of green spaces thanks to park forests. In Budapest, there is an average of 25 $m^2$ of forested parkland and only 6 $m^2$ of public park or garden per inhabitant. There is uneven use of green spaces, with well-positioned parks with cultural and historical value being constantly overcrowded [35].

Cemeteries play a dominant role in the urban structure due to their large surface area and, as a result of their function, their typically high proportion of green space. In addition to their memorial function, cemeteries represent a significant green space value, their green spaces being a key element of the capital's green space system [36] (Figure 7).

Both functioning and closed cemeteries are an important part of the capital's green infrastructure network. Cemeteries play an important role in the green infrastructure network of Budapest both because of their size (467 ha) and because of their high proportion of green space. The proportion of green space in cemeteries is well above the 40% defined in the OTÉK (Országos Településrendezési és Építési Követelmények, the National Town Planning and Building Requirements), even if the vegetation on the graves is not taken into account. The National Cemetery on Fiumei Road is the second-largest cemetery in Budapest, with the highest green cover (67%) [37]. For the cemeteries in operation, the Green Space Intensity Value ranges from 58% (Budafok cemetery) to 91% (New Public Cemetery), which is very high for the city. This value is largely influenced by the large mature trees in the cemeteries. Both the proportion of green areas and the ZFI value are higher for closed cemeteries; however, in many cases they are affected by the presence of weed species due to lack of maintenance, which negatively affects their use and value.

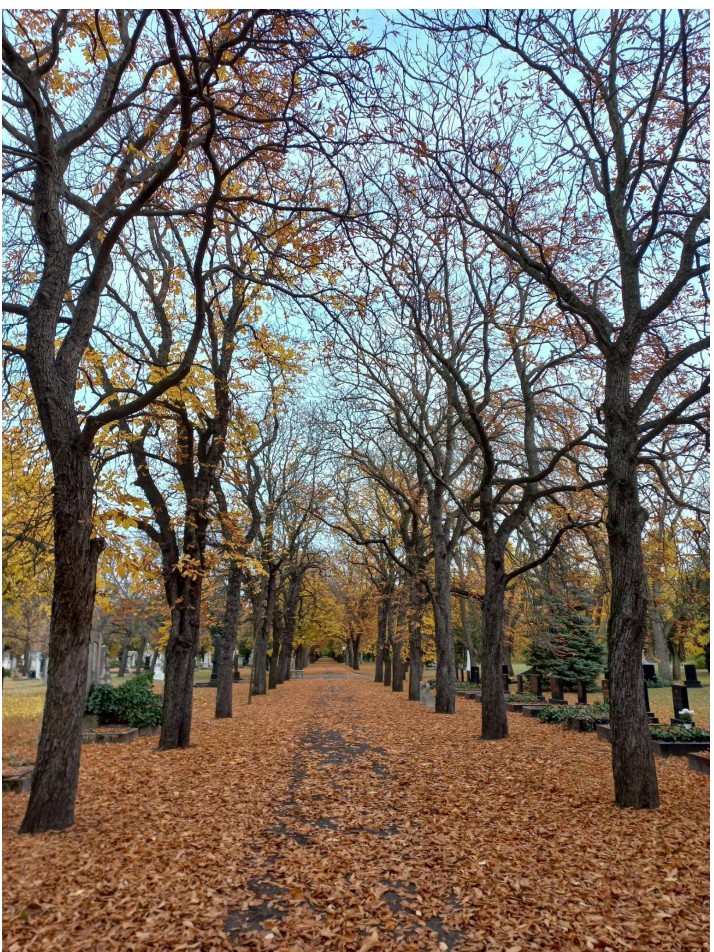

**Figure 7.** Chestnut tree alley in the National Graveyard on Fiumei Road (Photo: Imola Gecséné Tar, 2021).

Green spaces in cemeteries provide a range of ecosystem services to the population, such as improving air quality, enhancing the local climate, and providing aesthetic and recreational value [38]. In Hungary, the recreational use of cemeteries is not considered conventional; however, many people take advantage of the recreational opportunities offered by cemeteries. In addition to commemorative structures, almost all cemeteries have valuable mature tree species, tree lines, and rich biodiversity, which makes cemeteries an important part of biodiversity conservation [39,40]. In many cemeteries, both cemetery managers and visitors are already making serious efforts to preserve biodiversity; for example, in Farkasréti cemetery, bird feeders and bird boxes have been installed in several places (Figure 8).

The use of cemeteries as public green spaces contributes to the green coverage of the capital. The Action Area X of the Radó Dezső Plan, approved in 2021, concerns the cemeteries of Budapest, with the objective of "Protection, quality renewal and continuous professional maintenance of green infrastructure; development of an ecological approach to burial; efficient development and co-planning of public cemeteries; recreational use of functioning cemeteries; public use of closed cemeteries; temporary use of disused cemeteries as green spaces." The main tasks related to the achievement of these objectives, which are relevant to the present research, are "Better use of the recreational potential of existing cemeteries; public use of closed cemeteries; temporary green infrastructure use of disused cemeteries" and "Promotion of alternative burial methods; designation of urban areas suitable for forest burial" [35] (p. 62). The Capital intends to complete the planned improvements to the cemeteries by 2027.

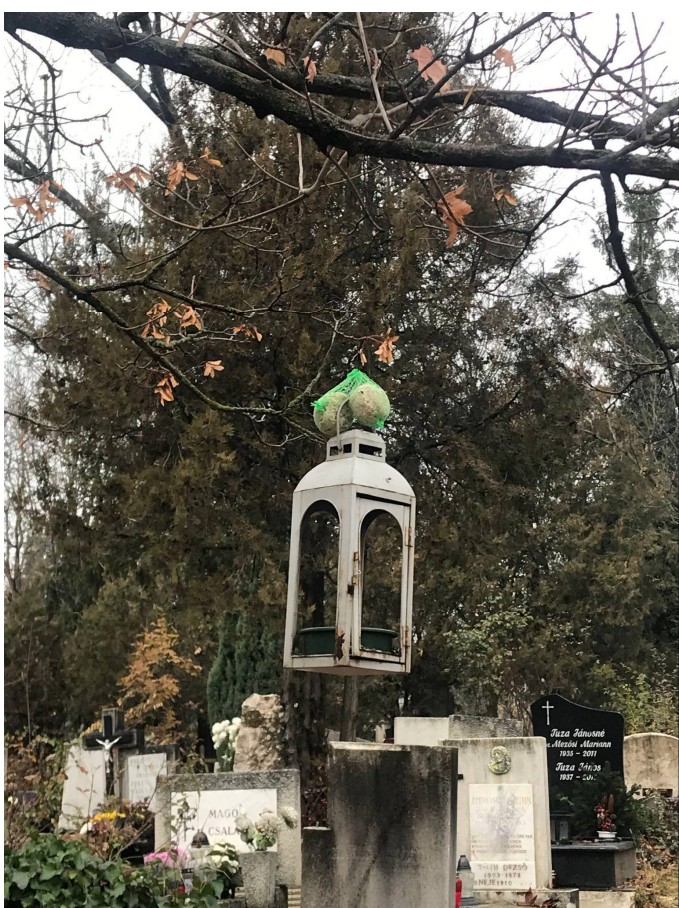

**Figure 8.** "Bird feeder" in Farkasréti cemetery (Photo: Ágnes Sallay, 2021).

### 1.4. Touristic Activities in Cemeteries

For thousands of years, cemeteries have had the fundamental role of providing a burial place for the deceased and a place of remembrance for the bereaved. This primary function has been complemented in recent decades by secondary and tertiary functions such as providing adequate green spaces for local climate, recreation, and tourism [41]. This expansion of functions has implied a number of conflicts; how can all these traditional and new functions be managed in parallel? How can the primary needs be fully satisfied alongside the secondary and tertiary functions? Can the duty of remembering the deceased persons in cemeteries be reconciled with tourism? In many cases, the human need to say a dignified farewell to the dead has been eroded during the COVID-19 pandemic, traumatizing society and increasing the demand for cemetery use. The lockdowns due to the pandemic have affected the way cemeteries are used, with many people finding cemeteries a suitably quiet place to relax and remember their loved ones; even if they were not buried in the cemetery, they could visit in proximity. For those cemeteries where the green assets and vegetation were well established, the cemetery has become a place to meet nature (Figure 6).

In many places throughout Western Europe, the daily recreational use of cemeteries by the local population has become a general need and constant demand; in addition to the traditional walking activities in cemeteries, other forms of exercise such as running (in many places with separate tracks available on site), silent sports such as yoga, and dog walking are allowed under regulated conditions.

Among the tertiary touristic functions, guided walks are the most widely accepted, with visitors introduced to the values and stories of the cemeteries, mainly built heritage and grave marker interpretation. The natural or green values of cemeteries are rarely highlighted; rather, the guides include ideas about the flora and fauna of the cemeteries in

other guided walks, for example in Highgate cemetery. Other tourism-related uses have been raised and implemented in different cemeteries; theater performances and concerts are more and more commonly organized, although these activities divide society and often provoke opposition from the local population, for whom the cemetery is a place of remembrance. There are various guided tours in the largest cemeteries of Budapest, where visitors can learn about the local history, the gravestones, and the famous people who rest in these cemeteries (Table 2). However, there are only a minority of guided tours that interpret the cemeteries as green spaces, presenting their living flora, and there are very few guided walks especially offered for children.

**Table 2.** The topics of guided walks in the largest cemeteries of Budapest (author's table with use of [42–44] sources).

| Organizing Institute or Company | The Name or the Theme of Walks and Guided Tours |
| --- | --- |
| Sétaműhely Ltd. | Trees and headstones—guided walks in the Farkasréti Cemetery<br>On the trails of Kohanites |
| Imagine Budapest<br>(Sétapálca Ltd.) | A past encased in stones—a tour of the Salgótarjáni Street Jewish Cemetery |
| Nemzeti Örökség Intézete (NÖRI)<br>National Heritage Institute | Parcels of artists<br>"Faster, Higher, Stronger"—Olympic medalists in the National Graveyard<br>"On hidden pathways"—cycling tour in the National Graveyard on Fiumei Road<br>Celebrated gipsy violinist<br>First World War—Military tombs<br>19th century's prime ministers<br>Musician souls<br>The artists of the "Nyugat" generation<br>Under the spell of "Thália"<br>The masters of Hungarian painting<br>The death cult of the socialism<br>The secrets of the plot 301—What do the graves tell us?<br>The Prime Ministers of the 20th century<br>Inventors and engineers<br>In the footsteps of great travelers<br>The gravestone artworks of Béla Lajta in the Jewish cemetery on Salgótarjáni Street<br>Budapest, Budapest, so wonderful<br>The life of an artist in sculptures<br>Sacred depictions in funerary art of tombs<br>In the footsteps of the Piarist Fathers and their disciples<br>Mausoleums and the National Graveyard on Fiumei Road<br>Those doomed to oblivion—Specific interpretation walk<br>Graves in the garden—plants and symbols in the cemetery<br>"That soul in me..."—Literary walk in the cemetery<br>Immortal art—Sculptors and sculptures<br>Remembering 1848–49<br>Ferenc Deák and his generation<br>Memento '56<br>History of the Jewish cemetery on Salgótarjáni street<br>Women, Muses, Destinies<br>Women on the trail of success<br>Gastro moods<br>Family walk with the little ones<br>Go, Hungarians! Athletes and sports leaders<br>Irregular school lessons in the cemetery<br>Secrets of an oasis in the heart of the city—The National Graveyard on Fiumei Road<br>(guided walk in English language) |
| Budapesti Temetkezési<br>Intézet (BTI)<br>Budapest Funeral Institute | Guided walks in the Farkasréti, Óbudai and Rákoskeresztúri Cemeteries |

The touristic use of cemeteries in Western Europe was already evident in the early 2000s. The Südwest-Kirchhof in Stahnsdorf, in the immediate vicinity of Berlin, is one of the first representatives of a landscape-style forestry cemetery, and the third largest in Germany. The cemetery, which was almost abandoned at the end of the Second World War, was the subject of a strategic plan, a programme drawn up in the early 2000s by the cemetery's advocacy association. After recognising the cemetery's cultural, historical, and artistic importance along with its natural value, the primary objective was to bring it back into public consciousness while introducing new ways of usage. Attracting community programmes, frequent invitations of the press, and a widespread sensitisation and awareness to unique burial opportunities have brought the result that the cemetery has now been reborn. It has become a popular destination for Berliners, as have, the cemetery gardens in the city center, which are operating as active elements in the green space infrastructure as a place for recreation. The function of the funeral chapel has been extended to include cultural functions such as conferences, exhibitions, and concerts. The cemetery employs volunteers and trainees who contribute to the success of the actions through their maintenance, research, and guide work. The cemetery's management has discovered and uses the opportunities offered by the media and the internet; periodic press conferences on the cemetery's cultural and historical significance, its artistic values, its current events, and even a promotional film have all been regularly created [11,45].

In terms of touristic use, the Central Cemetery of Vienna (Zentralfiedhof Wien) is a particularly successful example. In addition to the traditional use of the cemetery for burial and commemoration purposes, a number of improvements with diverse goals have been made in recent years:

- Classical music and rock concerts have been organized
- Sports facilities were developed, e.g., a running track was arranged in the cemetery,
- and electric bicycles for hire and carriage rides are offered as new services
- Regular night-time opening during the month of October to give visitors a night-time view of the cemetery
- Music walks offered, with live music accompanying the walkers
- Painting course offered, where visitors can create their own artwork souvenir of the cemetery
- A cemetery museum has been opened where visitors can learn about the history of the cemetery and the famous people buried there
- Nature-focused guided walks and camps for children
- A café and gift shop have been opened at the cemetery gate

The cemetery has responded to emerging digital needs as well; anyone can take part in a self-guided walk with a QR code (Hearonymus programme), and if required, visitors can find out information about the available burial places from a digital database. Visitors can even open a digital memorial link for a deceased loved one, with the possibility of storing commemoration materials, photos, and videos about them.

All these improvements to the Central Cemetery of Vienna have been carried out in close and regular consultation with the public, taking into account their opinions and requests, to the great satisfaction of the people of Vienna as shown by the fact that the cemetery was awarded the "best place in Vienna prize" in 2020 [46].

Following foreign trends, alternative ways of using cemeteries have already appeared in Hungary. The best example of this is the National Graveyard (Fiumei Road Cemetery), where visitors can take part in numerous thematic walks which introduce them to the stories of emblematic persons buried in the cemetery and of other people with interesting background histories. A more 'sporting' form of visit is exists where tourists can sign up for a bike tour of the cemetery. In this largest cemetery of Budapest, it is common to see mums walking with baby strollers, young people picnicking or studying, or sportsmen exercising, running, or cycling. Because the cemetery was designed and executed as a large green park area from the beginning of its creation and is planted with many valuable trees and alleys, the recreational popularity is not surprising. The urban wildlife established in

this cemetery is quite rich, and the site is comparable to a large-scale arboretum with its 56 hectares. It is important to mention the Memorial Museum operating in the National Graveyard, where visitors can learn about different aspects of Hungarian cemetery culture and burial traditions through permanent and temporary exhibitions [13].

## 2. Materials and Methods

This research focuses on the cemeteries of Budapest, analyzing them based on data collected during site visits and a research process employing relevant photograph and map sources. The theoretical foundations were established through literature sources on the following topics: (1) cemeteries–recreation and cemeteries–tourism relationship; (2) green spaces–green space infrastructure and the history of cemeteries in Budapest; (3) tourism activities in cemeteries–opportunities and foreign models. The main questions of our study were then formulated as follows: (1) what are the motivations of visitors for visiting cemeteries, and what are the values of each cemetery related to visitor motivations? and (2) what are the cemetery visiting habits and needs of the Hungarian population?

To answer the first question, we used literature sources to prepare an aggregation of cemetery visitor motivations followed by a survey of the attractions associated with these motivations in each cemetery. Visiting a cemetery can be either the main purpose of a trip or a secondary purpose, as a complementary part of it. As cemeteries have a wide range of attractive aspects, there are many motivations for visiting them; using the work of Tomašević (2018) [47], Moreno (2018) [48] and Pécsek (2015) [49] as a starting point, we defined the following motivation categories based on literature research and our own experience:

(1) Basic motivation: visiting family graves. This includes visiting, searching one's own or others' family roots and genealogy. A place of confrontation with mortality.

(2) Cultural motivation: the cemetery as an open-air museum, a repository of material and spiritual treasures.

 (i) Graves of famous people (kings, presidents, politicians, artists, writers, poets, actors, scientists, etc.)

 (ii) Architectural curiosities (emblematic constructions, crypts, mausoleums, etc.)

 (iii) Sculptural curiosities

 (iv) Places of interest, or a place of personal interest (e.g., the cemetery is featured in a film or a book), ideal for practicing a hobby (e.g., photography, painting)

 (v) A "must see" place, i.e., it is on the World Heritage List or on Tripadvisor

(3) National sentiment: visiting memorial parks, military cemeteries; participating on different on-site memorial events

(4) Admiration/love of nature: visiting cemeteries as other parks and arboretums. Special botanical and garden architecture interests, or more generally, a reason for escaping from urban pressures and providing relaxation and recreation

(5) Education and research: visits driven by an interest in history and culture. Visitors especially come to learn more about the history of the cemetery, the place where it was established, and the people who are buried there

(6) Religious motivation (pilgrimage): visiting graves of significant religious/ecclesiastical personalities (e.g., popes, church leaders) or tombs of religious and/or historical significance (e.g., the tomb of Jesus in Jerusalem). A place of confrontation with mortality

(7) The cemetery is an open-air leisure site with organized programs; visiting a concert or a performance in a cemetery

(8) No motivation: unplanned visits. No specific interest in the special features and offerings of the cemeteries. Curiosity, "let's see what we can find here" attitude

To answer the second question, we conducted a questionnaire survey in September–November 2021 among the population of Hungary with the intention of uncovering their cemetery visiting habits. In Hungary, the number of visits to cemeteries associated with All Saints' Day and Day of the Dead is high [50]; therefore, the questionnaire was distributed during this

period. We hoped this recent experience would increase the response rate and lead to more accurate answers. The questionnaire was completed in electronic form and was available online at https://forms.gle/PaZcbcRFidn4G1wKA (accessed on 1 November 2021). We wanted to know how often people visit cemeteries and for what reasons. We asked what activities not related to the basic memorial functions of cemeteries respondents considered to be acceptable.

The structure of the questionnaire can be found in the Appendix B. The questionnaire consisted of certain main questions and diverse supplementary questions. The main questions directly addressed the research topic, while the supplementary questions were asked in order to increase the reliability of the information obtained. For this reason, these latter were general demographic questions used to determine gender, age, education, etc. The questionnaire included open, semi-closed, and closed questions. In the case of the closed questions, respondents were asked to tick one of the pre-selected options. Semi-closed questions included an 'other option' category next to the offered ones, which allowed respondents to complete the list themselves if they did not find a suitable pre-written answer(s). Another example of a semi-closed question is where, in addition to yes/no answers, respondents were asked for the reasons behind their choice. For open questions, respondents were given the opportunity to formulate their answers in their own words.

The sampling was completely random. No attempt was made to limit or influence completion or to narrow down the pool of respondents. The aim was to obtain as wide a range of ages and interests as possible filling out the questionnaire and to determine the overall aspects of the typical visit. Due to the COVID-19 pandemic, we distributed the questionnaire online through professional and personal platforms, websites, and social media. The survey was successful, as 213 people completed it. Our objective was achieved in that the respondents were diverse in terms of gender, age, place of residence, education, and employment background.

In order to increase the reliability of the survey, another questionnaire was included for those who organized or led visits to cemeteries (Appendix C). This was used as a control, as some of the questions overlapped with the questionnaire for the general public. From these responses we gained better insight into the opportunities and problems from a tourism-specific point of view.

## 3. Results

With appropriate promotion, urban cemeteries could become an integral part of urban green tourism, as has already been the case for cemeteries abroad, e.g., in Paris and Prague. Visitors to urban cemeteries spend relatively long periods of time, even several hours, at these sites. The offered leisure activities, such as walking, photography, etc., contribute to a positive visitor experience. The traditional authenticity of cemeteries in terms of their extension, spatial structure, architecture, landscaping, artistic value, and local historical background, is sufficient to maintain their tourist appeal and visitors do not expect major improvements to infrastructure or attractions. Thus, from the visitor point of view, cemeteries do not require any major investment to enhance their attractiveness other than promotion of their highlights and conservation of their existing value (Figure 9).

### 3.1. Results of Research on Visitor Motivations

Visiting a cemetery might be the main purpose of traveling or a part of regular touring, or a primary or secondary motive for traveling, and might be driven by emotions of variable intensity. During our research we identified eight motivating elements in visitors to Budapest cemeteries. We identified the potential tourist attractions within each motivation category; then, through written sources, maps, and field visits, we identified which cemeteries in Budapest have which tourist attractions (Table 3).

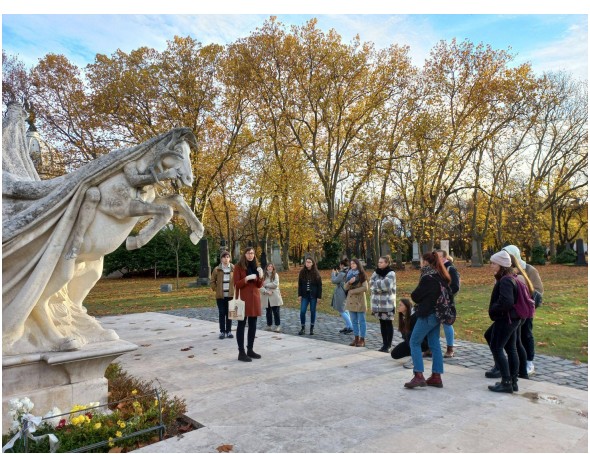

**Figure 9.** Guided walk in the National Graveyard on Fiumei Road (Photo: Imola Gecséné Tar, 2021).

**Table 3.** Visitor motivations, with related attractions and values in the cemeteries of Budapest (edited by authors, 2021).

| Visitor Motivations | Attractions | Representative Cemeteries |
|---|---|---|
| 1. Basic motivation | tombs and crypts of families | All functioning cemeteries |
| 2. Cultural motivation | | |
| 2.1. graves of famous people | graves of kings, presidents, politicians, artists, writers, poets, actors, scientists, etc. | National Graveyard on Fiumei Road<br>Farkasréti Cemetery<br>New Public Cemetery in Rákoskeresztúr<br>Farkasrét Jewish Cemetery<br>Óbudai Jewish Cemetery<br>Kozma Street Jewish Cemetery<br>Salgótarjáni Street Jewish Cemetery |
| 2.2. architectural curiosities | emblematic constructions, crypts, mausoleums, etc. | National Graveyard on Fiumei Road<br>Farkasréti Cemetery<br>Kozma Street Jewish Cemetery<br>Salgótarjáni Street Jewish Cemetery<br>Budafoki temető |
| 2.3. sculptural curiosities | cross, statue, carving | National Graveyard on Fiumei Road<br>Farkasréti Cemetery<br>Óbudai Cemetery<br>Salgótarjáni Street Jewish Cemetery<br>Óbuda Jewish Cemetery<br>Budafoki Cemetery<br>Kisszentmihályi Cemetery |
| 2.4. places of interest, or a place of personal interest, ideal for practicing a hobby (e.g., photography, painting) | the cemetery is featured in a film or a book | |
| 2.5. a "must see" place | it is on the World Heritage List or on Tripadvisor | |
| 3. National sentiment | memorial parks, military cemeteries graves being attached to wars, monuments of wars | National Graveyard on Fiumei Road<br>New Public Cemetery in Rákoskeresztúr |
| 4. Admiration/love of nature | Special botanical and botany or zoology value | National Graveyard on Fiumei Road<br>Farkasréti Cemetery<br>New Public Cemetery in Rákoskeresztúr<br>Óbudai Cemetery |
| 5. Education and research | curiosity of settlement history | National Graveyard on Fiumei Road<br>Salgótarjáni Street Jewish Cemetery<br>Closed cemeteries<br>outskirt, earlier communal cemeteries |
| 6. Religious motivation (pilgrimage) | tombs of religious and/or historical significance, calvary | All Jewish Cemetery<br>National Graveyard on Fiumei Road |
| 7. Local | organised programs, concert, or a performance | National Graveyard on Fiumei Road |
| 8. No motivation | | All functioning cemeteries |

It is important for cemetery managers, travel agency staff, and tourism organizations to use standardized classification categories in order to target and reach visitors with the right interests. An important role in promotion should be TripAdvisor ranking; the high position of cemeteries as places to visit in several destinations proves the significance of cemeteries as tourist attractions. Cemeteries should make a proper and thorough analysis of all possibilities in order to increase their attractiveness and accessibility as well as awareness about their cultural, historical and natural importance. Further research could be aimed towards practical implementation of the cemetery concept as a tourist attraction with a focus on travel agents and local authorities and on enhancing their understanding of cemeteries as tourist products which can enable them to create new programs and find new markets.

*3.2. Results of Questionnaire Survey*

The results of the two questionnaire surveys are presented separately in the following subsections.

3.2.1. Results of Visitor Questionnaire Survey

The majority of respondents were aged between 25 and 60 years (25–40 years 21.6%, 40–50 years 31.5%, 50–60 years 26.8%), female (83.6%), with a university/college degree (84%) and living in Budapest (54.9%) or in the Budapest agglomeration (14.6%). According to our survey, 37.1% of respondents visited a cemetery only once a year and 15.5% even less frequently; 29.1% visited quarterly, and 13.6% monthly.

First, we wanted to know whether the frequency of respondents' visits had changed due to the pandemic closures. For the majority (73.2%), there was no change; 23.9% visited cemeteries less frequently, and only 2.8% visited more often. Those who went to the cemetery more often than before were asked why. They were given the option of indicating more than one answer and were given the option of giving an individual answer; 32% expected to find peace of mind from their visits, 28% increased their knowledge by visiting the cemetery, 24% were able to walk and move around the cemetery at a reasonable distance during the closure period, and 18% had a cultural programme.

The purpose of the visit was explored from several perspectives: (1) Why do people usually visit cemeteries? and (2) Do they visit cemeteries during their travels? In both cases, several answers were possible, and individual answers were possible as well. The answers were quite varied. The primary purpose of the visit was "to visit the graves of my loved ones/relatives", at 81.7%; 21% and 27% of respondents, respectively, chose "to remember the deceased" (not necessarily in the cemetery where they are buried), "to visit the graves of famous people" (Figure 10), "to see the statues and buildings in the cemetery", and "to walk around, to relax". Individual responses included an interest in cemetery culture, walking the dog, chestnut/peanut picking, learning about the history of the village, peace and quiet, and greenery.

Of the respondents, 35.7% did not visit cemeteries during domestic trips. The rest (based on multiple choice) mainly visited the graves of relatives and friends (36.6%). Visiting historical sites (28.2%) and the graves of famous people (24.4%) were popular as well. The interest in visiting works of art (17.4%) and vegetation (17.8%) in the cemetery were almost the same, while 10.8% said they visited buildings/construction. Individual responses included "just looking around", "for the atmosphere" (tranquility + green), "to visit cemeteries that are unique/special in some way", "for cultural purposes", "to learn about local history", "as a tourist attraction", and "for the gravestones/inscriptions".

Fewer people visited cemeteries when traveling abroad, with 44.6% of respondents not visiting cemeteries. The majority of people who visited cemeteries abroad went to historical sites (35.2%) or to the graves of people they know (26.3%). Individual responses were dominated by the desire to learn about the cemetery culture of other countries and to visit unique/special/noteworthy cemeteries, as well as by the desire for green space and a peaceful atmosphere.

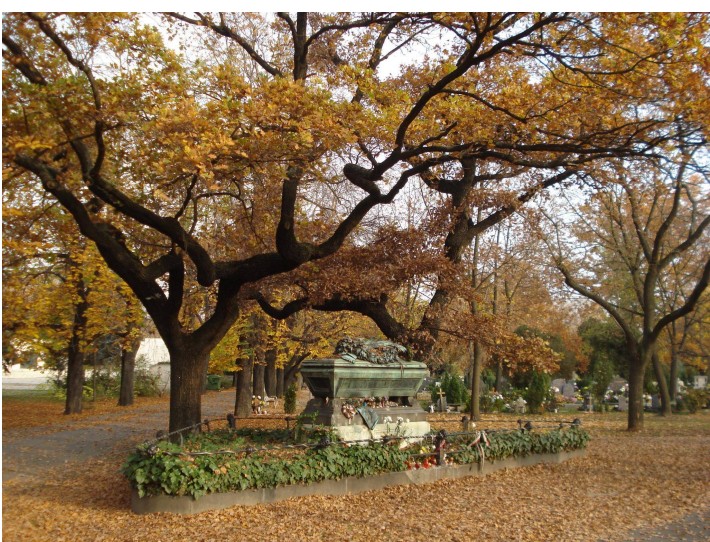

**Figure 10.** The tomb of János Arany in the National Graveyard on Fiumei Road (Photo: Imola Gecséné Tar, 2021).

Of the respondents, 40.8% had visited a cemetery for recreation. In addition to walking (74%) and sightseeing (74%), reading (5.2%) was the most common activity (multiple choice). Individual responses varied widely in their choice of activities, including photography (statues or Halloween), sports (running, cycling), guided walks, birdwatching, and concerts.

Of the survey respondents, 31.9% had attended a cemetery-related event. Based on individual responses, these includes any kind of commemoration (national holiday, World War II, Day of the Dead), wreath-laying, and guided walks. An equal number of respondents attended theater events and concerts. Only 10.8% participated in an event unrelated to the basic function of the cemetery. These included interesting events such as a photo lessons, drawing, nature walks, "the night of nightingales", book readings, exhibitions, and concerts, of which the latter two were the most frequently mentioned.

Well-kept cemeteries are important to visitors (92%), and this influences the time they spend there (65.7%). In terms of vegetation, the presence of trees/woods is considered the most important (93%, based on multiple choice), while shrubs (61.5%), grassy areas (60.6%) and flower beds (53.5%) play a significant role.

To conclude the questionnaire, we asked respondents what conditions they thought should be met in order for a cemetery to be suitable for recreation. Several answers could be ticked, and individual comments were possible. Of the answers we offered, the highest-ranked was green space (79.3%), followed by good accessibility (57.3%), open space between graves (57.3%), and a closed cemetery (11.7%). Among the individual responses, several respondents said that they did not consider cemeteries suitable for recreation. However, there were many constructive suggestions. Respondents felt it was important that recreational activities should not in any way interfere with the primary function of the cemetery and those who go there to commemorate the deceased. Several respondents mentioned basic infrastructure elements such as water supply, toilets, lighting, benches, waste bins, and security. They stressed the importance of being close to nature and the view, tidiness, well-kept graves, and the existence of quality green spaces. The creation of a park-like greenspace and memorial garden as a recreational area could be an important aspect of cemetery planning in the future. However, the first and most important thing to do is to change attitudes.

### 3.2.2. Results of Guide Questionnaire Survey

The questionnaire for tour guides was filled in by seven respondents, most of them contracted guides for the managing companies of Budapest's cemeteries (NÖRI and BTI), which provide regularly-conducted guided walks in the cemeteries they manage (namely,

National Graveyard on Fiumei Road, Jewish Cemetery on Salgótarjáni Road, Farkasréti Cemetery, Óbuda Cemetery, and Budafok Cemetery). The responses clearly showed that guided tours of cemeteries are in increasing demand among Budapest residents and tourists alike. Visitors go on guided walks mainly due to the cultural aspects of graves and memorials of famous people buried there and the stories associated with them, and additional attractions such as artistic sculptures and buildings make for a very charming experience. The wildlife, both flora and fauna, in the cemeteries are an additional attraction that adds color to the walks. During touristic use of cemeteries, the operators pay particular attention to ensuring that the main functions of the cemeteries are not compromised; thus, guided walks are organized during periods when burials are not in operation. During the guided walks, several companies use equipment such as earphones to provide adequate audibility while ensuring that loud speech does not disturb people visiting and caring for the graves of their loved ones.

Cemetery guides declare that the tourist potential of these facilities depends largely on the maintenance level of the green spaces which, especially during the growing season, provides an attractive background for the graves; in the summer, the shade of the trees makes the walks more pleasant for visitors. The development of appropriate infrastructure is essential for touristic exploitation; accessibility, parking, and restrooms are needed in sufficient quantities and quality. In the local metropolitan context of Budapest, a need for gift shops and catering facilities has not yet been formulated by visitors.

In addition to the primary functions of cemeteries, professional guides consider thematic walks and cultural activities (e.g., exhibitions, concerts) as possible secondary uses for memorial parks and graveyards.

## 4. Discussion

The responses to the two questionnaires show that people in Hungary are not yet open enough to attending activities in cemeteries other than attending and caring for the graves of their deceased relatives or commemorating honorable persons. This is somewhat contradicted by the fact that many people and a wide variety of programmes have taken part in cemeteries, which shows that people are not closed to the possibility. A change in attitude, as mentioned by one of the respondents, may be a reason for that, as we may currently be in an early phase of slow transformation. However, the variety of organized programmes listed in the questionnaire responses shows that cemetery managers are open to new potentialities. The Budapest Funeral Institute is explicitly open to challenges, although they are aware that changes should be made slowly and steadily, as sudden reforms can have the opposite effect [51]. A free small train tour has been organized in the Fiumei cemetery to complement the already-existing irregular school lessons. While the Day of the Bells took time to become established, people gradually accepted the idea of alternating childrens' programmes with light music and instrumental concerts in a cemetery. The Fiumei út Cemetery is a popular place for people to walk, push a pram, play ball with their children, or just lie on the grass with a blanket. It is home to the Museum of Funerary Art, Hungary's only specialized collection of funerary and memorial art. The crematorium in Csömöri Graveyard has a café where relatives can 'wait' for their loved ones to be buried in peace.

Our research clearly shows that the advanced types of cemetery tourism forms in Hungary belong today to the unconventional category. The touristic use of cemeteries is a non-conventional tourism industry sector because cemetery visitors mostly visit graves without using registered tourism services, but they still strengthen the tourism service industry by travelling to the site and using other services near the cemetery. Meanwhile, the responses indicate that this issue should be discussed and convenient options and solutions acceptable to all parties should be explored.

Nevertheless, we can state that there exists real cemetery tourism in Hungary. Not only do cemeteries in the capital increasingly appearing on the list of tourist attractions; those in the countryside with an interesting history and local graves of famous Hungarian

people do as well. According to Gábor Móczár, Director General of the National Heritage Institute (NÖRI), the main activity of facilities such as the National Graveyard is no longer to provide burial services; it is rather to present the diverse cultural values and promote dignified remembrance. For foreigners, the biggest attraction in cemetery tourism might be exploring and enjoying the local works of art. In June 2021, the NÖRI joined the Association of European Significant Cemeteries, a European network of historic and thematic cemeteries that includes the two cemeteries they manage (the National Graveyard on Fiumei Road and the Jewish Cemetery on Salgótarjáni Road). At the same time, the managing institution has joined the European Cemeteries Route, a more operational European cooperation community in terms of mutual travel programs, funding, and professional contacts, which has put Hungary on the map of European cemetery tourism in this respect [52]. A free app called FiumeiGuide has been developed which, in addition to presenting 150 priority highlights in the graveyard, currently allows users to plan their own personalized thematic tour.

The answers to the main questions of our study were provided by the capabilities and constraints identified during our research on the touristic use of Budapest cemeteries.

(1) In terms of tourist attractions and sights, the surveyed cemeteries are well-endowed. However, there is an urgent need to collect and identify these assets and values as thematic tourist attractions and to make their relevant information available to visitors. It is important to bear in mind that the cemeteries were not created for tourism purposes. They have gradually become tourist attractions throughout history. Thus, their primary function should not be negatively affected by their use for tourism and recreation. Although cemeteries are not yet conventional touristic destinations in Hungary, they have the potential to become more attractive in the future; their natural, built, and spiritual value and the tangible and intangible heritage they involve reinforce each other. Thus, cemeteries can be identified as open-air recreational areas with several attractions, where the past and present of a settlement as well as its development can be explored in a relatively small area during an individual or guided walking tour. Another beneficial factor is that most municipal cemeteries are open for a wide range of hours and are usually free of charge for visits. Attractiveness can be further enhanced by improving organized cultural programs and events on site. Depending on the cemetery, this could be the development of an existing attraction or the creation of a new one by unique or complex product improvement. This latter may include singular interventions as enhancements of the immediate surroundings around a main attraction, as well as any green space development.

(2) We have found a number of shortcomings in the tourist infrastructure of the cemeteries surveyed on the basis of both our personal site visits and the questionnaire responses. Tourism infrastructure in this case refers to the whole of the welcoming facilities, i.e., those infrastructural elements that can be used by the general public and that enable touristic activities. The improvement of basic tourism services and their background infrastructure is essential and covers, in particular, the development of utilities and pathway systems in a graveyard. This is important for the needs of both tourists and everyday visitors. For recreational and touristic purposes, it is important to provide the essential conditions for a longer stay. For example, there is a fundamental need to provide more benches and basic services such as restrooms. Improving accessibility in general is usually linked to complex touristic and infrastructural development, and is certainly a critical issue. The development of other services that serve mainly tourists, such as souvenir shops, should be preferably developed only after basic service needs have been satisfied.

(3) There are major gaps in the use of visitor management tools. In the case of historic cemeteries, reconciling their original function with touristic uses requires due diligence. Problems related to touristic use can be avoided or solved by using appropriate management methods. For example, problems are often caused by lack of infrastructure (toilets, cleaning facilities, catering facilities), resulting in pollution

and litter issues, or by tourists/visitors disturbing mourners and commemorators with their constant presence, talking, or general loudness. Solutions could include the preparation of visitor management tools, planning the movement of visitors in advance, better managing the potential negative impact of visits, and avoidance or mitigatation of negative consequences. Passive exhibition techniques appropriate to the character of the cemetery may be good practice for visitor education, although in most cemeteries only an overview map of the cemetery layout is displayed. There are rarely any signs or printed materials (brochures, guides, and more detailed special publications) providing information on the specific values and attractions of the cemeteries and visualizing their locations. For larger cemeteries, it is necessary to develop thematic itineraries for guided tours or self-exploration walks. During self-guided tours, visitors can follow a pre-edited brochure along the way.

## 5. Conclusions

Based on our research, we can conclude that cemeteries, as green open spaces in major cities, play a highly-important role in urban greenspace systems and infrastructure such as local climate control, daily recreation, and tourism services. Cemeteries provide profoundly essential ecosystem services for both the residents of surrounding urban areas and for visitors. Among the ecosystem services provided by cemeteries, their regulating services have a main role in the adjustment of climate issues, while their cultural services contribute to the preservation of traditional and cultural heritage aspects as well as to the recreational and touristic function of these sites.

The green surfaces of cemeteries are traditionally defined by tree alleys which delimit the pathways separating the burial plots, as well as by grassland areas and flower beds between the graves; these are, as a whole, important for visitors. As the vegetation becomes older and larger in well-established and maintained sites, the shading effect of trees becomes increasingly important, as they provide tolerable urban climate conditions for visitors in the summer. Large cemeteries with mature trees can have temperatures during the summer several degrees lower than in the surrounding areas, providing a pleasant resting place for visitors. Our questionnaire research clearly shows that the level of maintenance and care for the vegetation in cemeteries is essential for both visitors and tourist guides, two target groups whose opinions must not be neglected.

Despite the recognised green value of urban cemeteries, the presentation of their semi-natural vegetation and habitat is unfortunately little-developed in Hungary, although this could help to diversify guided walks and even enlarge environmental education functions in these cultural complexes. However, any tourist development should prioritize the fundamental needs of the local population, namely, unhindered commemoration and care of graves. Hopefully, in the short term and with the right development accompanied by public education, remarkable cemeteries with significant green spaces can play a satisfactory role in future green tourism in urban areas in Hungary.

The use of cemeteries for tourist purposes must remain in balance with their traditional memorial function. We have examined the possible tourist exploitation of different types of cemeteries (e.g., closed/inactive or active cemeteries, gridded, and landscape-style cemeteries) within an ethical framework (Figure 7). We have found that in addition to their architectural and artistic value, the green surface intensity of a cemetery largely contributes to their wider potential for tourism uses.

The greenspace potential of a cemetery consists of several factors. If the vegetation cover and plant use are adequate, the ecological importance of the area is evident. However, only cemeteries with a sufficient size and maintenance level are suitable for other green space functions such as recreation purposes. Cemeteries with a smaller area may be suitable for daytime recreation, while cemeteries with a larger surface area can be visited from a greater distance, similar to urban public parks. It is important to note here that the primary memorial function of cemeteries should not be compromised, which is why closed cemeteries are more suitable for recreation than functioning cemeteries. It is important to mention that, even in a closed cemetery, recreational activities should be allowed only after

due consideration. Walking, reading, and quiet contemplation do not disturb a functioning cemetery; however, certain sports activities and noisy programmes that attract large crowds are not even conceivable in a closed cemetery. Some non-intensive sports (e.g., running, yoga) or cultural events that attract fewer people and do not result in damage either physically or spiritually may be allowed in a closed cemetery.

The recreational and touristic use of cemeteries are partly interconnected, as the green potential of a cemetery determines the interest in the cemetery (e.g., visits for botanical values). In terms of tourist use, priority should be given to those cemeteries in Budapest that can already offer other attractions in addition to their primary function. A cemetery is most attractive for tourism if it has suitable potential beyond its original function, i.e., visitors seek out its special architectural, natural, or artistic values, the burial places of famous people, or sites of cultural and historical interest.

**Author Contributions:** Conceptualization, Á.S. and ZS.M.; methodology, Á.S. and ZS.M.; writing—original draft preparation, Á.S., ZS.M., I.G.T. and K.T.; validation, Á.S., ZS.M. and I.G.T.; formal analysis, K.T.; investigation, Á.S., ZS.M. and I.G.T.; resources, Á.S., ZS.M. and I.G.T.; data curation, ZS.M.; writing—review and editing, Á.S., ZS.M., I.G.T. and K.T.; visualization, Á.S., ZS.M. and I.G.T.; supervision, Á.S.; project administration, Á.S. All authors have read and agreed to the published version of the manuscript.

**Funding:** The dissemination of research outcomes was partly supported by The Hungarian Tourism Association Foundation (MTSZ).

**Institutional Review Board Statement:** Not applicable.

**Informed Consent Statement:** Not applicable.

**Data Availability Statement:** Not applicable.

**Conflicts of Interest:** The authors declare no conflict of interest.

## Abbreviations

| | |
|---|---|
| NÖRI | Nemzeti Örökség Intézete—National Heritage Institute |
| BTI | Budapesti Temetkezési Intézet—Budapest Funeral Institute |
| MAZSIHISZ | Magyarországi Zsidó Hitközségek Szövetsége—Federation of Hungarian Jewish Communities |

## Appendix A

**Table A1.** Operating cemeteries in Budapest: an overview table (by authors, 2021).

| Operating Public Cemeteries | | | |
|---|---|---|---|
| **Name of the Cemetery** | **Location in Budapest** | **Cemetery Management** | **Features** |
| Cemetery in Óbuda (Óbudai temető) | 3rd district | BTI | guided tours |
| Tamás Street Urn Cemetery (Tamás utcai urnatemető) | 3rd district | BTI | urn cemetery on the site of the former Békásmegyeri cemetery |
| Megyeri Cemetery (Megyeri temető) | 4th district | BTI | extension of the former village cemetery |
| National Graveyard on Fiumei Road = Kerepesi cemetery (Fiumei úti sírkert = Kerepesi temető) | 8th district | NÖRI | national pantheon, guided tours |
| New Public Cemetery in Rákoskeresztúr (Rákoskeresztúri Új Köztemető) | 10th district | BTI | the largest cemetery in Hungary, guided tours |
| Farkasréti Cemetery (Farkasréti temető) | 11th district | BTI | the largest cemetery in Buda, guided tours |
| Cemetery in Rákospalota (Rákospalotai temető) | 15th district | BTI | extension of a former village cemetery |
| Cemetery in Cinkota (Cinkotai temető) | 16th district | BTI | extension of a former village cemetery |

**Table A1.** *Cont.*

| Operating Public Cemeteries | | | |
|---|---|---|---|
| **Name of the Cemetery** | **Location in Budapest** | **Cemetery Management** | **Features** |
| Cemetery in Pestszentlőrinc (Pestszentlőrinci temető) | 18th district | BTI | extension of a former village cemetery |
| Old Cemetery in Kispest (Kispesti Öreg temető) | 19th district | BTI | reopened former village cemetery |
| Cemetery in Kispest (Kispesti temető) | 19th district | BTI | extension of a former village cemetery |
| Cemetery in Pesterzsébet (Pesterzsébeti temető) | 20th district | BTI | extension of a former village cemetery |
| Cemetery in Csepel (Csepeli temető) | 21st district | BTI | extension of a former village cemetery |
| Cemetery in Budafok (Budafoki temető) | 22nd district | BTI | former village cemetery |
| Angeli Road Urn Cemetery (Angeli úti urnatemető) | 22nd district | BTI | urn cemetery on the site of the former Nagytétényi cemetery |
| Operating Jewish cemeteries | | | |
| Name of the cemetery | Location in Budapest | Cemetery management | Features |
| Jewish Cemetery in Óbuda (Óbudai izraelita temető) | 3rd district | MAZSIHISZ | |
| Jewish Cemetery in Kozma Street (Kozma utcai izraelita temető) | 10th district | MAZSIHISZ | the largest Jewish cemetery in Hungary |
| Orthodox Jewish Cemetery in Gránátos Street (Gránátos utcai ortodox izraelita temető) | 10th district | MAZSIHISZ | |
| Farkasréti Jewish Cemetery (Farkasréti izraelita temető) | 11th district | MAZSIHISZ | |

**Appendix B**

Questionnaire on the use of cemeteries (visitor questionnaire)
I. Questions about the respondent:

- age: 18–25 y, 25–40 y, 40–50 y, 50–60 y, 50–70 y, 70 over
- sex: female/male
- education: 8 primary school, secondary school, university/college
- place of residence: Budapest, Budapest agglomeration, large city, medium-sized city, small town, village, other options to define:

II. Questions on cemetery visiting habits:

- How often do you visit a cemetery?
- weekly
- several times a month
- on a quarterly basis
- yearly
- less frequently

III. Has the frequency of your cemetery visits changed due to the pandemic closures?

- yes, I've been there less often
- yes, I have gone there more often
- No, it has not changed

IV. If you went to the cemetery more often during the closures, what was the reason? (multiple answers are possible)

- I expected spiritual comfort from the visits

- It was possible to walk and move around the cemetery at a reasonable distance during the closures
- The visit to the cemetery provided a cultural programme
- I expanded my knowledge with what I saw in the cemetery
- other options to define:

V. What is the purpose of your visit to a cemetery? (multiple answers are possible)

- visit the graves of my loved ones/relatives
- remember the deceased (not necessarily in the cemetery where they are buried)
- visit the graves of famous people
- get to know statues, artworks and buildings in cemeteries
- for walking and relaxing

VI. Do you visit cemeteries when you travel in your home country? (you can tick more than one answer)

- yes, I mainly visit the graves of relatives and friends
- yes, I visit cemeteries as historical sites
- yes, I visit the graves of famous people
- yes, I visit artworks and statues in cemeteries
- yes, I visit buildings and other architectural elements
- yes, I am interested in the green surface and vegetation of cemeteries (protected and special species)
- yes, other options to define:
- no

VII. Do you visit cemeteries when you travel abroad? (you can tick more than one answer)

- yes, I mainly visit the graves of relatives and friends
- yes, I visit cemeteries as historical sites
- yes, I visit the graves of famous people
- yes, I visit artworks and statues in cemeteries
- yes, I visit buildings and other architectural elements
- yes, I am interested in the green surface and vegetation of cemeteries (protected and special species)
- yes, other options to define:
- no

VIII. Have you ever visited a cemetery for recreational purposes?

- yes
- no

IX. If yes, what were you doing there? (multiple answers are possible)

- walking
- reading
- contemplating
- doing sport activity (e.g., running)
- other options to define:

X. Have you ever participated in a cemetery-related event (e.g., commemoration, other ceremony)?

- yes
- no

If yes, what was it?
XI. Have you ever participated in an event not related to the cemetery's core function (e.g., concert, exhibition, bird ringing activity, etc.)?

- yes
- no

  If yes, what was it?
  XII. Is the maintenance of the cemetery important to you?

- yes
- no

  XIII. Is the time spent in the cemetery influenced by the current physical condition of the cemetery or by the abundance of green spaces there?

- yes
- no

  XIV. What kind of vegetation do you consider important in a cemetery, apart from the plants on the graves?

- trees, alleys
- flower beds
- shrubs
- grasslands

  XV. What conditions do you think must be provided by a cemetery to be attractive for recreation? (multiple answers are possible)

- good accessibility
- it should be a closed cemetery (with no more active burials)
- free space between graves
- good coverage of green areas
- other options to define:

**Appendix C**

Questionnaire for tourist guides on the use of cemeteries
I. In which cemetery do you usually organize guided tours?

- National Graveyard on Fiumei Road (Fiumei úti temető/Kerepesi úti temető)
- Farkasréti Cemetery (Farkasréti temető)
- Jewish Cemetery in Salgótarjáni Road (Salgótarjáni úti zsidó temető)
- other options to define:

  II. How often do you organize guided walks?

- weekly
- once a month
- several times in a month
- a few times a year

  III. Do you organize guided walks specifically dedicated to the cemetery/ies or as part of a more complex tourist programme?

- I organize themed cemetery walks on different sites
- I also bring tourists to cemeteries on my sightseeing walks.
- I organise walks only in the National Graveyard on Fiumei Road (Fiumei úti temető/Kerepesi úti temető)
- I organise walks only in the Farkasréti Cemetery (Farkasréti temető)
- I organise walks only in the Jewish Cemetery on Salgótarjáni Road (Salgótarjáni úti zsidó temető)
- other options to define:

  IV. How many people visit cemeteries with you each year?

- Under 100 people
- Between 100–200 people

- Between 200–500 people
- More than 500 people

V. What age group of people usually takes part in your organized walks?

- mostly aged between 18–30
- mostly aged between 30–50
- mostly aged between 50–70
- mostly over 70

VI. What are the main elements you are showing during your walks? (multiple answers are possible)

- the graves of famous people
- the statues in the cemetery,
- notable buildings
- old/notable trees
- other options to define:

VII. During guided trips abroad, are you taking groups to a cemetery?

- yes, we visit historical sites
- yes, we visit the graves of famous people
- yes, we visit works of art and statues in cemeteries
- yes, we visit them for their architectural values (e.g., buildings)
- yes, I highlight the vegetation of the cemeteries (protected and special species)
- no, I don't have these kind of activity

VIII. Is the maintenance of a cemetery important for your interpretation as a guide?

- yes
- no

IX. What kind of vegetation do you consider important for guided walks in a cemetery, apart from the plants on the graves? (multiple answers are possible)

- trees, alleys
- flower beds
- shrubs
- grassland areas

X. Is the time spent in the cemetery being influenced by the current condition of the cemetery or by the amount of green space?

- yes
- no

XI. What services would be needed in cemeteries for guided groups? (multiple answers are possible)

- parking places
- restroom, toilet
- benches
- souvenir shop
- catering facilities
- other options to define:

XII. Would you organize any events in the cemeteries that are not strictly related to the primary function of the cemetery?

- yes
- no

XIII. If yes, what would it be?

- concert

- exhibition
- theatre performance
- other options to define:

  XIV. What conditions do you think must be provided by a cemetery to become a suitable place for recreation? (multiple answers are possible)

- having good accessibility
- being a closed cemetery
- having enough free spaces between graves
- having good coverage of green spaces
- other options to define:

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
