# Peer review of "Cemeteries as a Part of Green Infrastructure and Tourism"

_sustainability, doi:10.3390/su14052918_

Round 1

Reviewer 1 Report

The topic of the paper is interesting, relatively underrepresented in tourism research, and particularly relevant in the current Covid-period when the value of urban green spaces in recreation and tourism has increased significantly.

By offering an overview of good practices for alternative uses from European cemeteries, the paper has a markedly practical orientation and it may indeed be useful for developers in cities where such uses are not popular yet (in and outside of Hungary).

However, from a scientific point of view, the paper needs a stronger focus and a better structure.

The recreational potential and the tourism potential of a cemetery are two different concepts, and the authors should be continuously aware of this difference in their analysis.

Since the title of the paper highlights the theme of “green infrastructure” and one of the research aims is “to identify the green space values of Budapest's cemeteries besides their well-known cultural and architectural significance”, this topic should be highlighted in the theoretical background of the paper (describing the situation in Budapest is not enough). The theoretical part of the paper needs to be improved in other aspects, too: while the authors do not provide a thorough synthesis of the existing literature (of cemeteries as green spaces, as recreational spaces, as tourist spaces, etc.), the presentation of the cemeteries of Budapest is unnecessarily lengthy and exhaustive for a non-Hungarian reader. I suggest the authors to check out the works of (1) Venbrux, (2) Tanaś and (3) Dancausa Millán et al. on tourism in cemeteries, (4) Swensen et al. and (5) Quinton and Duinker on the recreational value of cemeteries, as well as (6) Ratz on All Saints’ Day-induced travel (actually, a Hungarian contribution to the literature which the authors might already be familiar with). Research findings are also mixed with the theoretical background or with the methodology (see Tables 1-3).

Figure 6 – the hierarchy of cemetery functions – should be substantiated by the literature (and primer/seconder/tercier should be corrected as primary/secondary/tertiary).

The research questions should be reformulated (although the paper is generally well written, it would benefit from proofreading), and the research methods should be explained in details, especially the first questionnaire survey (what was the survey population? what kind of random sampling technique was used? how was the questionnaire administered?). Selecting the Fortepan database as opposed to contemporary social media sites for the photo analysis should also be explained, since this decision has led to a past-oriented, almost historical approach that does not fit very well into the original rationale of the study.

            The discussion of the findings includes valuable ideas, particularly for practitioners. The authors should make an effort to better explain the contradictions in their results (e.g. “The responses to the questionnaires show that people in Hungary are not yet open enough to doing other activities in a cemetery than attending and caring for the graves of their deceased relatives or commemorating honourable persons. Although this is contradicted by the fact that according to the questionnaire survey, many respondents have already participated in a wide range of organised activities in metropolitan cemeteries.”).

Although the manuscript needs revision before publication in Sustainability, this is a valuable paper and I hope that the authors will opt for resubmission.

Author Response

Review 1 - answer
Dear Reviewer,
Thank you for your thorough work and the time you took to prepare this review. We thank you for your many positive comments, your praise and overall positive assessment of the study and your suggestions for further reflection. We agree with many of your points. The constructive critical comments have contributed to the higher quality of our article. Below, we respond to the comments, suggestions and critiques made in the review.

Our overall changes to the article:
The study has undergone a major structural revision. It has been given a stronger focus and deeper literary support. The sections on cemetery history, the green space scheme and the results of the photo analysis have been combined into a sub-chapter. In this way, the research findings from different perspectives complement and support each other, providing a sufficient basis for the discussion of the topic.

The study has been supplemented with a more in-depth literature review subsection summarising the findings of the sources we consider relevant to the relationship between tourism and recreation/green space, and tourism and cemeteries. 
Photos and illustrations have been moved and edited to reflect changes in content.
The research questions and results were reviewed and revised consistently. The research questions have been refined and a stronger, more transparent coherence between questions, methods, and results has been established. Accordingly, the Discussion and Conclusions chapters have been added to.
The bibliography has been carefully reviewed and any formal inaccuracies identified have been corrected.

Responding to the comments and questions that have been raised:
The proposed structural changes have been made.
In the theoretical background section, the role of cemeteries in recreation and tourism has been defined based on the appropriate literature. In the study we have sought to use the two concepts in a more conscious and distinct way. The suggested authors' works have been reviewed and the sources relevant to this paper have been used.
The history of Budapest's cemeteries has been shortened and contrasted with the development of the green space system. Photographic analysis has been used to support these historical sections. Thus, by combining these subsections, a complex, multi-perspective historical overview was produced, which provided a solid basis for subsequent analyses.

The research questions were reformulated and the link between the questions and the results was confirmed. The methodological description has been clarified and explained in more detail. 
To resolve this discrepancy, appropriate additions have been made to the results.
Figure 6 (renumbered Figure 4) has been relocated and given appropriate literature support.

Reviewer 2 Report

The article deals with an important topic. It is written in an interesting way.

Cemeteries are Memorial places and a special national and pan-European heritage. 

Figs. 6 rightly enumerates the functions of cemetery grounds. However, the indication of the importance of individual functions is debatable. Such a distinction would require research.

Of course, as the authors point out, these areas have additional advantages. There are interesting architectural and sculptural objects. There are complexes and objects of natural value. Therefore, they can be areas of interest for tourists or recreational areas.

Perhaps, however, it is worth emphasizing the limitations indicated in this respect. The Viennese example can serve in a certain sense not as a model, but also as a warning. The limit that cannot be crossed may be far away. According to the reviewer, it is difficult to agree with popular music concerts or sports competitions in cemeteries.

Such reservations were, in a way, found themselves in the answers in the surveys and in the authors' conclusions. Therefore, it is worth indicating precisely what possible events increasing the tourist attractiveness of cemeteries would be acceptable in the case of Budapest.

In this respect, it would be worthwhile to make arrangements. The presented survey results do not contribute any useful design information. The questions should focus on identifying very specific recommendations and reservations. This would make it possible to adapt the tasks to the mentality and demand of residents and tourists visiting Budapest's cemeteries. For example, in some European countries, it would probably not be acceptable to introduce dogs or ride bicycles outside a strictly defined route.

With the other postulates of the authors regarding the service facilities, the availability of information, it is difficult to disagree. However, it must be admitted that these are rather obvious issues.

The role of cemeteries in Budapest's urban green system is poorly presented. Indicating the surface itself is not enough. Also with regard to the role of cemeteries as natural resources, the conclusions regarding their tourist popularization are obvious and devoid of specific indications. It is worth paying attention to the specificity of these areas, the difference from the areas of parks. It would be interesting, for example, to decide whether greenery should be kept in a meticulous order or rather in a more naturally disordered way, whether isolation or a sense of security is valued more, etc.

Author Response

Dear Reviewer,
Thank you for your thorough work and the time you took to prepare this review. We thank you for your many positive comments, your praise and overall positive assessment of the study and your suggestions for further reflection. We agree with many of your points. The constructive critical comments have contributed to the higher quality of our article.Below we respond to the comments, suggestions and critiques made in the review.

Our overall changes to the article:
The study has undergone a major structural revision. It has been given a stronger focus and deeper literary support. The sections on cemetery history, the green space scheme and the results of the photo analysis have been combined into a sub-chapter. In this way, the research findings from different perspectives complement and support each other, providing a sufficient basis for the discussion of the topic.

The study has been supplemented with a more in-depth literature review subsection summarising the findings of the sources we consider relevant on the relationship between tourism and recreation/green space, and tourism and cemeteries. Photos and illustrations have been moved and edited to reflect changes in content.

The research questions and results were reviewed and revised consistently. The research questions have been refined and a stronger, more transparent coherence between questions, methods, and results has been established. Accordingly, the Discussion and Conclusions chapters have been added to.
The bibliography has been carefully reviewed and any formal inaccuracies identified have been corrected.

Responding to the comments and questions that have been raised:
We fully agree with the suggestion that tourism improvements to cemeteries can only be implemented in consultation with the concerned public if it enjoys their support.The reception and acceptance of developments may even vary from one cemetery to another, depending on the population concerned. In any case, a more detailed questionnaire survey and several rounds of consultation with the population are necessary before any concrete development proposals are drawn up. In this article, we wanted to show the possibilities that already exist abroad and that are worth considering in Budapest.

The maintenance of cemeteries and related security issues may indeed be relevant to development, but this is beyond the scope of this article.

Figure 6 (renumbered Figure 4) has been relocated and given appropriate literature support

Reviewer 3 Report

Dear author,

The topic in discussion is very actual and important, mainly in a moment where "Dark tourism" is a reallity in many destinations.

I believe that this article could be improved in some areas:

  • Discuss the "Dark tourism" as a trend where the Cemiteries are one of the areas visited by tourists;
  • Clarify the methodology in order to be more clear the different methods in use to collect data
  • Choose just 2 of the data collected for the article, you present many different things and this is confusing for readers
  • With the data collected you can do 2 different articles, one in the demand perspective and the other in the offer perpective
  • Review tables, some have errors or missing letters
  • Review References. The online references at the end of the list need to be changed.
  • Look for the information of Sustainability article rules in order to check the info for references
  • Discussion of results and Conclusions need to be improved

Author Response

Dear Reviewer,
Thank you for your thorough work and the time you took to prepare this review. We thank you for your many positive comments, your praise and overall positive assessment of the study and your suggestions for further reflection. We agree with many of your points. The constructive critical comments have contributed to the higher quality of our article.
Below we respond to the comments, suggestions and critiques made in the review.

Our overall changes to the article:
The study has undergone a major structural revision. It has been given a stronger focus and deeper literary support. The sections on cemetery history, the green space scheme and the results of the photo analysis have been combined into a sub-chapter. In this way, the research findings from different perspectives complement and support each other, providing a sufficient basis for the discussion of the topic.

The study has been supplemented with a more in-depth literature review subsection summarising the findings of the sources we consider relevant to the relationship between tourism and recreation/green space, and tourism and cemeteries. 

Photos and illustrations have been moved and edited to reflect changes in content.

The research questions and results were reviewed and revised consistently. The research questions have been refined and a stronger, more transparent coherence between questions, methods, and results has been established.

Accordingly, the Discussion and Conclusions chapters have been added to.
The bibliography has been carefully reviewed and any formal inaccuracies identified have been corrected.

Responding to the comments and questions that have been raised:
Dark tourism is intentionally not discussed in detail in this article, but its importance is mentioned in the introduction.

References have been adapted to the journal's standards

Round 2

Reviewer 2 Report

The article has been significantly improved. It still does not contain answers to the most important detailed questions for project decisions. Probably obtaining such information was beyond the reach of the Authors' capabilities. However, the text presents a lot of valuable information, which is why it is worth publishing.

Reviewer 3 Report

Dear authors,

Congratulations for this new version of the article.

The improvements are really good and it seems that it is a "new" article.

I believe that the results presented have interest for the academic community.